# Frequency-Dependent Dielectric Permittivity and Water Permeability in Ordered Mesoporous Silica-Grafted Fluorinated Polyimides

**DOI:** 10.3390/polym16121716

**Published:** 2024-06-16

**Authors:** Jaemin Son, Hwon Park, Minju Kim, Jae Hui Park, Ki-Ho Nam, Jin-Seok Bae

**Affiliations:** Department of Textile System Engineering, Kyungpook National University, Daegu 41566, Republic of Korea; woals4272@knu.ac.kr (J.S.); sd9854@naver.com (H.P.); qminjup@knu.ac.kr (M.K.); kah142@knu.ac.kr (J.H.P.)

**Keywords:** fluorinated polyimide, amino-functionalized mesoporous silica, hydrophobic cross-linked network, dielectric constant, water permeability

## Abstract

Polymers with a low dielectric constant (*D*_k_) are promising materials for high-speed communication networks, which demand exceptional thermal stability, ultralow *D*_k_ and dissipation factor, and minimum moisture absorption. In this paper, we prepared a series of novel low-*D*_k_ polyimide films containing an MCM-41-type amino-functionalized mesoporous silica (AMS) via in situ polymerization and subsequent thermal imidization and investigated their morphologies, thermal properties, frequency-dependent dielectric behaviors, and water permeabilities. Incorporating 6 wt.% AMS reduced the *D*_k_ at 1 MHz from 2.91 of the pristine fluorinated polyimide (FPI) to 2.67 of the AMS-grafted FPI (FPI-*g*-AMS), attributed to the free volume and low polarizability of fluorine moieties in the backbone and the incorporation of air voids within the mesoporous AMS particles. The FPI-*g*-AMS films presented a stable dissipation factor across a wide frequency range. Introducing a silane coupling agent increased the hydrophobicity of AMS surfaces, which inhibited the approaching of the water molecules, avoiding the hydrolysis of Si–O–Si bonds of the AMS pore walls. The increased tortuosity caused by the AMS particles also reduced water permeability. All the FPI-*g*-AMS films displayed excellent thermooxidative/thermomechanical stability, including a high 5% weight loss temperature (>531 °C), char residue at 800 °C (>51%), and glass transition temperature (>300 °C).

## 1. Introduction

Recently, the remarkable progress of electronic devices capable of high-speed data communication has further increased the demand for electronic substrates with a low dielectric constant (*D*_k_) and high electrical insulation properties [1,2,3]. Reducing the permittivity of materials with electrical insulation properties can improve signal transmission speed and efficiency, which is an effective way to promote the development of high-frequency and high-speed flexible circuit boards [4]. Aromatic polyimides (PIs) have been broadly used as insulating layers and electronic packages in the microelectronic industry and have been proposed as potential candidates for printed circuit boards, flexible display screens, and next-generation interlayer dielectric materials because of their excellent thermal stability, mechanical strength, dielectric properties, and chemical resistance [5,6]. However, conventional PIs have relatively high dielectric constant (*D*_k_) values, ranging from 3.0 to 3.5, owing to the polarization due to their polar imide groups [7,8]. Moreover, hydrophilic moieties, such as carbonyl and amine groups, facilitate the penetration of water molecules into PIs in the ambient atmosphere, leading to an increase in the dielectric constant and functional deterioration [9,10]. Generally, the water absorption rates of the aromatic PI are relatively high, at 2.51% [11], and other examples are 1.91% for 2,2-bis(4-cyanatophenyl)propane [12], 3.4% for CYCOM 977-2 epoxy resin [13], and 10% for polybenzimidazole (PBI) [14]. In contrast, the water absorption rates of poly(ethylene terephthalate) (PET) and poly(phenylene sulfide) (PPS) are relatively low, at 1.1% [15] and 0.026% [16], respectively. Despite its low percentage, water absorption has an obvious influence on the dielectric properties because water molecules have a much higher dielectric constant (*D*_k_ = 80) than PIs [5,17,18].

Reducing the density or polarizability are two effective strategies for minimizing the *D*_k_ of PIs. It was demonstrated that introducing bulky fluorinated substituents into the backbone can significantly reduce the *D*_k_ of PIs due to the strong electronegativity and low atomic polarizability of fluorine [19,20]. In addition, the introduction of fluorine endows PIs with good organo-solubility and lowers water absorption [21]. Nonetheless, fluorinated polyimides (FPIs) generally have poor thermomechanical stability and high chemical sensitivity, which are particularly troublesome for low-*D*_k_ materials. Under such circumstances, organic/inorganic hybrids have evolved as an effective alternative owing to their tunable physical properties provided the prudent selection of fillers.

Mesoporous silica, such as polyhedral oligomeric silsesquioxanes, Santa Barbara Amorphous-15, or Mobil Composition of Matter No. 41 (MCM-41), has emerged as a research hotspot for reducing *D*_k_ and improving other properties [22,23,24]. Among them, MCM-41 exhibits an intrinsically low *D*_k_ of 1.4–2.1 due to its regular arrangement of cylindrical mesopores (3–4 nm) that form a one-dimensional pore system. Furthermore, MCM-41 presents a unique opportunity for preparing organic/inorganic hybrids with an inorganic core containing Si–O–Si bonds and organic functional groups on the periphery, which can increase compatibility with the polymer matrix [25]. Against this background, the introduction of cross-linked MCM-41 could be an effective solution to reduce the *D*_k_ and enhance other indispensable properties, including water resistance and thermomechanical stability. The cross-linked structure of MCM-41 can effectively restrict the segmental mobility of FPI chains and prevent the molecular segments from being oriented along the external electric field direction, thus reducing the *D*_k_ of FPIs.

In this study, we prepared organic/inorganic hybrids of an FPI and amino-functionalized mesoporous silica MCM-41 (AMS) by in situ polymerization and examined their dielectric properties, including the permittivity and dissipation factor (*D*_f_). The *D*_k_ at 1 MHz decreased to an ultralow value of 2.67 with a systematic addition of the AMS, with a promisingly low *D*_f_. Furthermore, we investigated the influence of AMS particles on the water absorption and thermomechanical stability of the FPI.

## 2. Materials and Methods

### 2.1. Materials

Sodium silicate solution (~10.6% Na_2_O and ~26.5% SiO_2_), cetyltrimethylammonium bromide (CTAB, >98%), and (3-aminopropyl)triethoxysilane (APTES, >99%) were purchased from Sigma-Aldrich (Saint Louis, MO, USA); 4,4′-(hexafluoroisopropylidene) diphthalic anhydride (6FDA, >98.0%) and 4,4′-oxydianiline (ODA, >98%) were purchased from Tokyo Chemical Industry, Co., Ltd. (Tokyo, Japan); and 1-methyl-2-pyrrolidinone (NMP, >99%), toluene (≥99.5%), ethyl alcohol (>99%), acetic acid (>99%), and hydrochloric acid (HCl, 37%) were purchased from Duksan Chemical Co., Ltd. (Incheon, Republic of Korea). Deionized water was obtained from a Milli-Q Ultrapure water purification system (MilliporeSigma, Burlington, Massachusetts, USA). All the chemicals were used without further purification.

### 2.2. Synthesis of NH_2_-MCM-41

To synthesize NH_2_-MCM-41, 3.25 g of CTAB was dissolved in 38 mL of deionized water at 40 °C. Then, 13.9 g of sodium silicate solution was slowly added to the system to induce silicate condensation. Subsequently, the mixture was heated at 100 °C for 24 h under static conditions and then cooled for easy handling, after which its pH was adjusted to 10 with a 50% acetic acid solution. The reaction and pH adjustment were repeated two more times, followed by washing and filtering with deionized water. The product was placed in an oven at 100 °C for 24 h. The sample was subsequently washed and filtered three times using a solution containing 2.5 g of HCl in 100 mL of ethyl alcohol and then dried and calcined at 550 °C in a furnace to remove the CTAB template. The mixture containing 2.5 g of synthesized silica and 2.5 g of APTES was refluxed in 100 mL of toluene for 24 h to obtain the amino-functionalized mesoporous silica MCM-41. The product was washed and filtered three times with ethanol and dried in the oven.

### 2.3. Synthesis of the NH2-MCM-41-Grafted Fluorinated Polyimide Films

A measured quantity of NH_2_-MCM-41 (0, 0.5, 1, 3, and 6 wt.%) was dispersed in NMP by sonication for 30 min. Stoichiometric equivalent amounts of ODA (0.6 g, 3 mmol) and 6FDA (1.33 g, 3 mmol) were added into the NH_2_-MCM-41/NMP suspension. The resulting mixture (11–16 wt.% solid in anhydrous NMP) was mechanically stirred for 24 h in an N_2_ atmosphere to produce a viscous NH_2_-MCM-41-grafted fluorinated poly(amic acid) (FPAA) intermediate solution. Subsequently, the homogeneous solution was cast onto a clean glass substrate and thermally cured under a sequential temperature programming (90 °C/2 h, 150 °C/1 h, 200 °C/1 h, 250 °C/30 min, 300 °C/30 min, and 400 °C/30 min). Finally, the resultant NH_2_-MCM-41-grafted FPI films were self-stripped from the glass substrate by soaking in deionized water and further dried at 80 °C for 24 h in a convection oven.

### 2.4. Measurements

Transmission electron microscopy (TEM) was performed using a Titan G2 80–200 instrument (FEI, Hillsboro, OR, USA) with ChemiSTEM technology. Scanning electron microscopy (SEM) with energy-dispersive X-ray spectroscopy (EDS, Ultim Extreme, Oxford Instruments, Abingdon, Oxfordshire, UK) was performed using an SU8220 microscope (Hitachi, Tokyo, Japan) at an acceleration voltage of 10 kV. Fourier-transform infrared (FTIR) spectra were obtained with a Nicolet iS5 (Thermo Fisher Scientific Inc., Waltham, MA, USA). X-ray diffraction (XRD) was conducted on an Empyrean X-ray diffractometer (Malvern Panalytical Ltd, Malvern, UK) with Cu Kα radiation (λ = 1.54 Å). Thermogravimetric analysis (TGA) was conducted with an SDT 650 (TA Instruments, New Castle, DE, USA) under N_2_ flow at a heating rate of 20 °C/min. Dynamic mechanical analysis (DMA) was conducted with a DMA 850 (TA Instruments) at a heating rate of 3 °C/min with a load frequency of 1 Hz in air. Dielectric properties were measured by a 4294A impedance analyzer (Agilent, Santa Clara, CA, USA) with a 16451B dielectric test fixture (Agilent, Santa Clara, CA, USA) in the frequency range of 40 Hz to 30 MHz at 25 °C. The dielectric tests were performed following the ASTM D150 standard [26]. The dielectric constant (ε′) was calculated as per Equation (1).
(1)$$\varepsilon^{\prime }=\frac{C_{m}}{\varepsilon_{o}}\frac{t}{A}=\frac{C_{m}\cdot t}{\varepsilon_{0}\times \pi \times {\left(d/2\right)}^{2}}$$
where ε_0_ is the vacuum permittivity (8.85·10^−12^ F m^−1^), *C*_m_ is the measured capacitance, *A* is the electrode area, *d* is the electrode diameter, and *t* is the film thickness.

Water vapor transmission rates (WVTRs) were measured using a PERMATRAN-W 3/61 (Ametek Mocon, Brooklyn Park, Minnesota, USA) instrument at 38 °C and relative humidity of 100%, following the ASTM F-1249 standard [27]. The WVTR data were normalized to the film thickness to obtain the water vapor permeability (WVP) using Equation (2).
(2)$$\mathrm{WVP}=\frac{\mathrm{WVTR}}{P\cdot (R_{1}-R_{2})}\cdot l$$
where *l* is the average film thickness, *P* is the water vapor pressure at 38 °C, and *R*_1_ and *R*_2_ represent the moisture gradients.

Surface wettability was analyzed by measuring the Young–Laplace’s contact angle using the sessile drop method at ambient temperature (~23 °C) in the air by Phoenix300 (SEO, Suwon, Republic of Korea). The contact angle was recorded with a droplet volume of 3 μL after 3 s from droplet deposition, with at least five replicate samples. For each reported value, all the measurements taken from different surface locations were averaged, and the standard deviation was calculated.

## 3. Results and Discussion

The AMS was synthesized via the hydrolysis and condensation of silicate precursors in the presence of cationic surfactants under basic conditions, followed by the post-grafting of organic grapes, as shown in Figure 1a. The TEM micrograph of the AMS particles demonstrates a parallel channel-like porous structure with a regular hexagonal arrangement, which is representative of MCM-41 (Figure 1b). The surface modification of the AMS seems to hardly affect the hexagonal porous structure of MCM-41. The XRD patterns of the non-functionalized mesoporous silica (MS) and AMS are displayed in Figure 1c. Three well-resolved diffraction peaks of (100), (110), and (200) suggest good crystallinity with the ordered hexagonal mesoporous architecture of MS [28,29]. The small diffraction peaks of (110) and (200) were not found in the XRD pattern of AMS, indicating that –NH_2_ groups on the AMS surface increased the structural disorder. However, the original honeycomb-patterned mesoporous scaffold was not destroyed during the post-grafting amination. Thermogravimetric weight loss curves for MS and AMS are shown in Figure 1d. The TGA curve of the AMS exhibits three weight loss intervals that are distinct from MS: (i) desorption of the water linked to the silica surface in the range of 50–150 °C, (ii) fragmentation of the organic functionalities attached to the AMS surface in the range of 200–600 °C, and (iii) mesoporous structure disruption above 600 °C [24,30].

Figure 2 shows an overview of the synthesis scheme for the FPI-*g*-AMS films via in situ polymerization. To achieve a high-molecular-weight FPI, the FPAA intermediate precursor was synthesized using tetracarboxylic dianhydride 6FDA and ODA in a stoichiometric equivalent ratio. The FPAA chain grafting of AMS resulted from the reaction of the amino end group of the AMS with the anhydride-chain end group of the FPAA backbone. The FPAA-*g*-AMS was dissolved in NMP to form a viscous homogeneous solution. After bar-coating this solution onto a clean glass substrate, the imidization was performed by a step-cure process (90 °C/2 h, 150 °C/1 h, 200 °C/1 h, 250 °C/30 min, 300 °C/30 min, and 400 °C/30 min). The thickness of the cured FPI-*g*-AMS films was ca. 60 µm.

The FTIR spectra of the FPI-*g*-AMS films are presented in Figure 3a. The characteristic absorption peaks at 1784 cm^−1^ and 1718 cm^−1^ were observed for all samples, corresponding to asymmetric (imide I) and symmetric (imide II) stretching vibrations of C=O, respectively. The absorption peak near 1370 cm^−1^, associated with C–N stretching vibration (imide III), appeared after the thermal imidization of the FPAA-*g*-AMS precursor [31]. Meanwhile, the absorption peak at 1650 cm^−1^, assigned to amide carbonyl stretching vibration (amide I), disappeared, indicating the complete conversion of the amic acid into imide bonds [32]. The absorption bands arising from the –CF_3_ group of the 6FDA moiety are located in the region of 1320–1170 cm^−1^, which overlaps with the peak of the Si–O–Si asymmetric stretching vibration in the FPI-*g*-AMS films [33]. Additionally, the absorption peak resulting from the stretching vibration of Si–O–Si bonds in AMS around 1070 cm^−1^ is overlapped by the C–H stretching vibration band in the PI backbone [34,35]. Nonetheless, the peak intensity of Si–O–Si stretching was slightly enhanced with the increasing AMS loading, confirming that the AMS is present in the resulting FPI-*g*-AMS films.

The FPI-*g*-AMS films exhibited broad halos in the range of 10–30° (2θ), which is attributed to their amorphous nature (Figure 3b). The average interchain distance (*d*-spacing) of all the samples was calculated from the position of the respective maxima using Bragg’s equation. The interchain *d*-spacing slightly increased from 5.78 Å for pristine FPI-*g*-AMS-0 to 5.85 Å for FPI-*g*-AMS-1, which can be attributed to the increased free volume and loose distribution of PI chains.

Figure 3c,d shows the SEM images of the cross-sectional morphologies of the FPI-*g*-AMS films. Pristine FPI-*g*-AMS-0 reveals a relatively smooth and flat fracture surface. However, the fracture surface roughness of the FPI-*g*-AMS films increased with AMS loading, suggesting that the AMS distorted the path of the crack tip, impeding crack propagation [36]. In addition, the SEM results show that AMS particles are distributed uniformly in the FPI matrix up to 3 wt.% AMS loading. For a highly loaded system (FPI-*g*-AMS-6), only a little agglomeration was observed in the FPI matrix. The EDX elemental mapping images further demonstrate that AMS particles are homogeneously dispersed in the FPI matrix (Figure 3e), confirming that amino-functionalization enhances the interfacial interaction between the fillers and the polymer chains.

The thermal degradation behaviors of the FPI-*g*-AMS films are shown in Figure 4a and Appendix A. It can be seen that the thermal degradation of all the samples occurs through a single stage in the temperature range of 100–800 °C, which indicates a favorable phase interconnection between the AMS and the FPI matrix. The pristine FPI-*g*-AMS-0 film started to decompose by 5% (*T*_d5%_) at 539 °C and retained 57% char residue at 800 °C. The *T*_d5%_ values of the FPI-*g*-AMS films were in the range of 531–536 °C, similar to the pristine FPI-*g*-AMS-0 film. However, the thermal degradation of the FPI-*g*-AMS films was accelerated above 600 °C, and char residue at 800 °C slightly decreased with increasing AMS loading. This could be ascribed to the relatively higher thermal conductivity of the mesoporous silica compared to that of the FPI matrix.

Figure 4b–d and Appendix A demonstrate the temperature dependence of the viscoelastic behaviors of the FPI-*g*-AMS films. The DMA curves for all the samples exhibit glass-to-rubber transitions. As shown in Figure 4b, all the samples had constant dynamic storage moduli (*E’*) before the glass transition temperature (*T*_g_), reflecting the solid-like properties of the rigid imide backbone. However, above the *T*_g_, the increase in *E’* with the addition of AMS is significant, which can be attributed to the good interfacial interaction between the AMS and the FPI matrix. Figure 4c,d show the loss factor (tan δ) of the FPI-*g*-AMS films. The pristine FPI-*g*-AMS-0 and FPI-*g*-AMS films exhibit similar *T*_g_ values regardless of the AMS loading. Nevertheless, as the AMS loading in the FPI-*g*-AMS films increases, the peak intensity of tan δ declines, as clearly shown in Figure 4d. These results suggest weak relaxation of FPI chains near the surface of the AMS around *T*_g_, due to strong filler–polymer interactions.

The frequency dependences of the *D*_k_ and *D*_f_ of the FPI-*g*-AMS films are shown in Figure 5 and Appendix A. The dielectric constants of all the samples declined slightly with increasing frequency. Typically, the *D*_k_ of polymeric materials gradually decreased as the frequency increased at constant temperature. This is because the orientation of the dipole moment was not fast enough to keep up with the oscillations of the applied alternating electric field at high frequencies. The dielectric constants of the FPI-*g*-AMS films at 1 MHz showed a conspicuous decline with increasing AMS loading, i.e., from 2.91 of the pristine FPI-*g*-AMS-0 film to 2.67 of FPI-*g*-AMS-6. The dielectric constant was closely related to molecular-orientation-dependent polarization. The well-defined mesoporous architecture of the FPI-*g*-AMS films considerably reduced the number of polarizing molecules per unit volume. On the other hand, the increase in interfacial polarization due to the high concentration of AMS (>3 wt.%) suppressed the reduction effect of the *D*_k_ in the FPI-*g*-AMS films. This polarization enabled the charged particles to overcome the energy loss caused by the thermal motion induced by the electric field. The dissipation factor quantifies the amount of energy absorbed or dissipated as heat within a dielectric material when subjected to an electric field. The *D*_f_ also exhibited frequency-dependent characteristics. At high frequencies, the rapid reversal of the polarization direction led to greater energy dissipation as heat, increasing the *D*_f_ of the polymer. Figure 5b shows that the FPI-*g*-AMS films had stable and low *D*_f_ values (~0.0026 at 1 MHz) over a wide frequency range, from 10^3^ Hz to 10 MHz. This observation is attributed to the rigid segmental dynamics of FPI chains grafted to AMS particles.

Water absorption is a major factor that degrades the electrical performance and reliability of low-*D*_k_ polymeric materials in electronic devices, increasing current leakage [37]. The water barrier properties of the FPI-*g*-AMS films were investigated using surface static water contact angle and WVP analysis. The experimental WVP data calculated from the WVTR was fit to the solution of Fick’s second law of diffusion in Figure 6b, assuming the water vapor concentration to be close to zero at the exit side of the films, as follows [38].
(3)$$P^{○}=P\sqrt{\left(\frac{4d^{2}}{\pi \mathrm{Dt}}\right)}\sum_{n=0}^{\infty }\exp\left[-\frac{d^{2}}{4\mathrm{Dt}}{(2n+1)}^{2}\right]$$
where *P*° = *Jd*/*p*, *P* = *J*_s_*d*/*p = SD/p*, *J* and *J*_s_ denote the water vapor flux at time *t* at steady state, *d* is the sample thickness, *p* is the differential water vapor pressure, *P* is permeability, *D* is diffusivity, and *S* is solubility.

The permeability, diffusivity, and solubility of the FPI-*g*-AMS films determined from the best fitting of the model calculations are summarized in Appendix A. According to the solution–diffusion model, condensable vapors could cause increased polymer chain mobility and plasticization, thus dissolving the rigidified polymer chains at the filler–polymer interface and diffusing into the nm-scale pores of the AMS particles. In addition, once the water vapor enters large-diameter pores, the diffusion rates can be increased dramatically. The WVP of the pristine FPI-*g*-AMS-0 film is 1.52 × 10^–8^ mol s^−1^ m^−1^ atm^−1^, as shown in Figure 6b and Appendix A. The addition of AMS significantly influenced the WVP values of the FPI-*g*-AMS films. The largest reduction in WVP (1.10 × 10^–8^ mol s^−1^ m^−1^ atm^−1^) was observed at 3 wt.% AMS. This could be attributed to the increase in the hydrophobicity of the AMS surfaces by APTES moieties, thus inhibiting the approaching of the water molecules and avoiding the hydrolysis of the Si–O–Si bonds of the AMS pore walls [39]. The increased tortuosity caused by the AMS particles also reduced water permeability. Even though they could diffuse into the AMS pores, the increase in the diffusion path lengths when they traversed the film led to reduced permeabilities. For FPI-*g*-AMS-6, because of the agglomerated large size of the AMS particles, the AMS surface could not be wrapped perfectly by the FPI chains, leading to the formation of interfacial voids and defects. The water vapor had a tendency to diffuse through these non-selective voids, resulting in higher permeability in FPI-*g*-AMS-6. These explanations clearly follow from the permeability, diffusivity, and solubility behaviors according to the AMS concentration, as shown in Figure 6b–d. The solubility behavior is similar to that of permeability, whereas the diffusivity of all the samples is nearly identical. This result indicates that when incorporating AMS particles into the FPI matrix, the difference in permeability is ascribed to the solubility rather than the pathway tortuosity effect. The surface static water contact angle also strongly depends on the AMS loading (Figure 6a). We observed that the contact angles of the FPI-*g*-AMS films were larger than those of the pristine FPI-*g*-AMS-0 film. However, at 3 wt.% AMS, the contact angle decreased due to the formation of polymer–filler interfacial voids and defects caused by AMS agglomeration. These results are consistent with the diffusion behavior of the water molecules, as shown in Figure 6b–d.

## 4. Conclusions

We successfully prepared novel low-*D*_k_ FPI films containing an MCM-41-type AMS, which exhibited an ultralow dielectric constant of ~2.69 and a very low dissipation factor of ~0.0026 at 1 MHz. The surface of the FPI-*g*-AMS films exhibited obvious hydrophobic characteristics that endowed the films with outstanding water resistance, with FPI-*g*-AMS-3 showing a remarkably low water permeability of 1.10·10^−8^ mol s^−1^ m^−1^ atm^−1^. This could be attributed to an increase in the hydrophobicity of AMS surfaces by the APTES moieties, thus inhibiting the approaching of the water molecules and avoiding the hydrolysis of the Si–O–Si bonds of the AMS pore walls. The increased tortuosity caused by the AMS particles also reduced water permeability. In addition, the FPI-*g*-AMS films showed high thermooxidative/thermomechanical stability, including a high 5% weight loss temperature (>531 °C), char residue at 800 °C (>51%), and glass transition temperature (>300 °C). Therefore, the introduction of a hydrophobic cross-linked organic/inorganic hybrid network is beneficial to effectively reduce the *D*_k_, improve water resistance, and simultaneously maintain the overall physical properties of PIs. The proposed FPI-*g*-AMS films showed great potential in the applications of next-generation dielectric materials in the microelectronic industry.

## Figures and Tables

**Figure 1 polymers-16-01716-f001:**
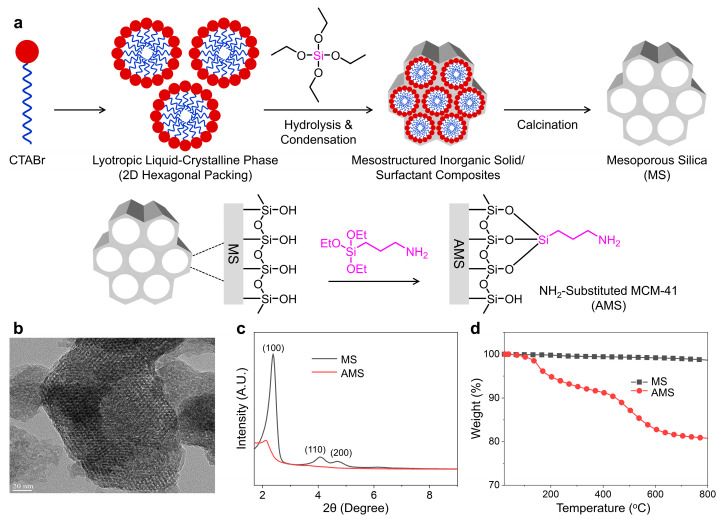
(**a**) Schematic illustration of amino-functionalized mesoporous silica (AMS) via the sol–gel reaction catalyzed in a basic medium and post-synthesis grafting method. (**b**) Transmission electron microscopy of AMS. (**c**) X-ray diffraction (XRD) patterns and (**d**) thermogravimetric analysis (TGA) curves of non-functionalized mesoporous silica (MS) and AMS.

**Figure 2 polymers-16-01716-f002:**
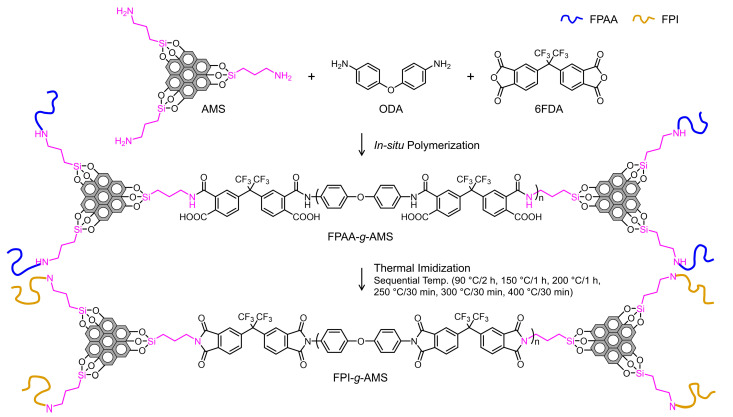
Synthesis of AMS-grafted fluorinated polyimide films.

**Figure 3 polymers-16-01716-f003:**
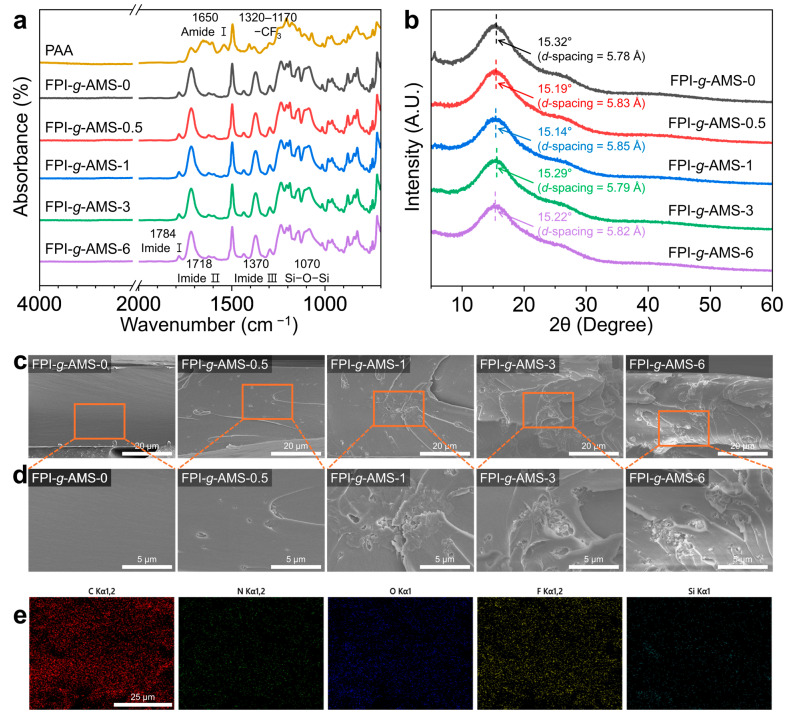
(**a**) Fourier-transform infrared spectra and (**b**) XRD patterns of the FPI-*g*-AMS films. (**c**) Low-magnification and (**d**) high-magnification SEM fracture surface morphologies of the FPI-*g*-AMS films. (**e**) Elemental mapping images of the FPI-*g*-AMS-3 film.

**Figure 4 polymers-16-01716-f004:**
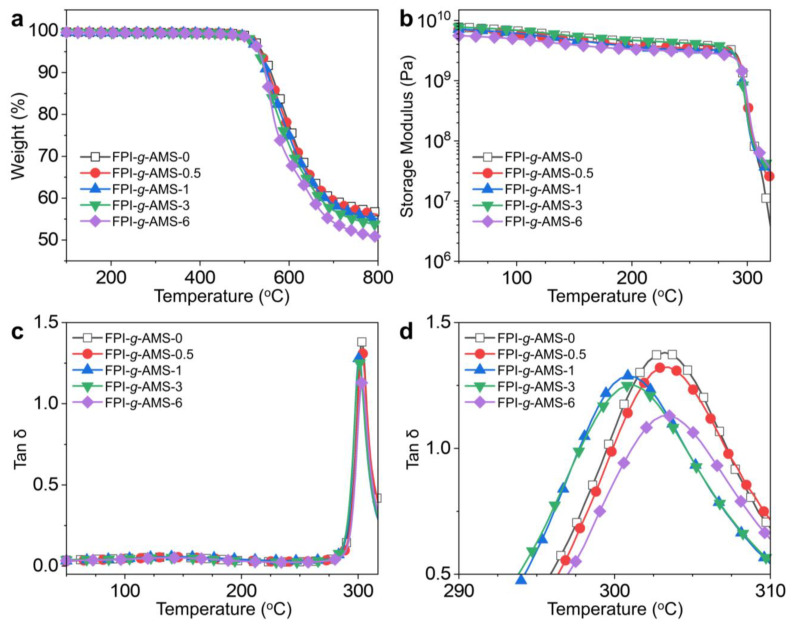
(**a**) TGA curves, (**b**) storage modulus, (**c**) tan δ, determined by dynamic mechanical analysis, and (**d**) magnification region of tan δ in the range of 290–310 °C for the FPI-*g*-AMS-3 film.

**Figure 5 polymers-16-01716-f005:**
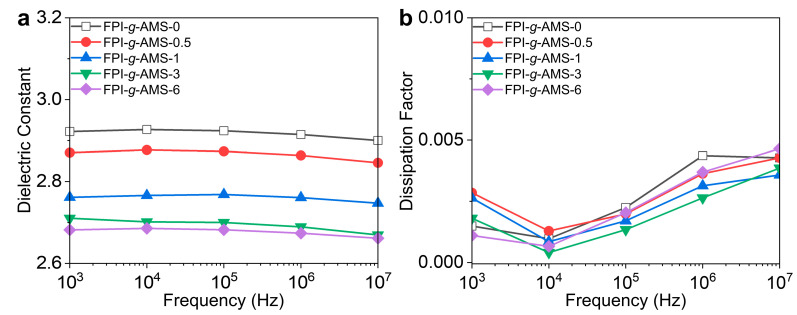
Frequency-dependent (**a**) dielectric constant (*D*_k_) and (**b**) dissipation factor (*D*_f_) of the FPI-*g*-AMS films at room temperature.

**Figure 6 polymers-16-01716-f006:**
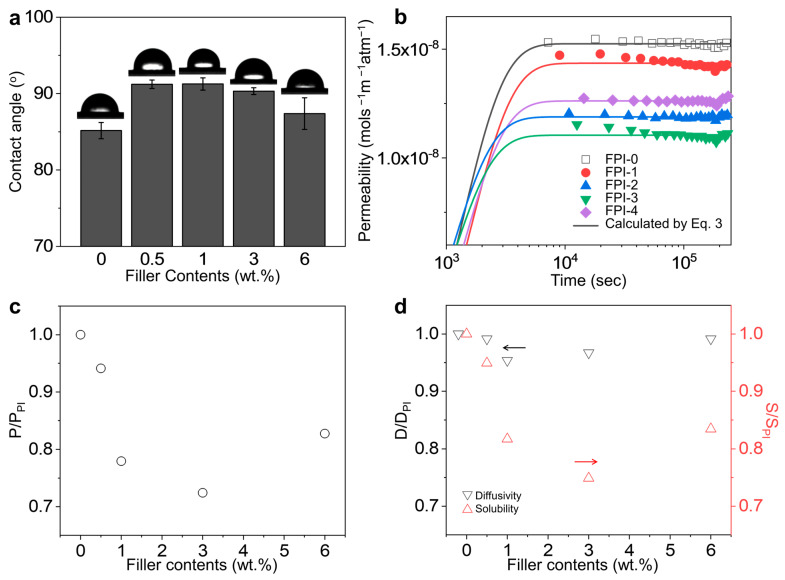
(**a**) Surface static water contact angles of the FPI-*g*-AMS films. (**b**) Water vapor permeability of the FPI-*g*-AMS films as a function of time compared to the model calculated by Equation (3). Filler effects on the FPI-*g*-AMS films: (**c**) permeability and (**d**) diffusivity and solubility. Accordingly, the y-axis for diffusivity is on the left in black, while the y-axis for solubility is on the right in red. Each arrow indicates the direction of its respective y-axis.

## Data Availability

The original contributions presented in the study are included in the article/Appendix A, further inquiries can be directed to the corresponding authors.

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
