# Peer review of "Frequency-Dependent Dielectric Permittivity and Water Permeability in Ordered Mesoporous Silica-Grafted Fluorinated Polyimides"

_polymers, 2024, doi:10.3390/polym16121716_

Round 1

Reviewer 1 Report

Comments and Suggestions for Authors

The dielectric measurement results could be enriched by presenting also ac conductivity.

corrections: line217 Figure 4 (not 6)

also at lines 261-264 please clarify solubility S calculation.

Author Response

  1. The dielectric measurement results could be enriched by presenting also ac conductivity.

Author reply: We appreciate the reviewer’s valuable comment. As semiconductors become more integrated, the interconnections between transistors can slow down. The RC delay, which characterized interconnection delay, can be improved by lowering the specific resistivity of the conductor or by lowering the dielectric constant of the insulating material. Thus, it is important to use dielectric materials with low dielectric constants to reduce signal delay. In addition, low dissipation factor values are required to improve signal transmission rate and prevent signal distortion due to dielectric loss. Besides to these low dielectric constants and dielectric losses, excellent thermal stability and low water absorption are also required for insulating layers. Currently, copper is commonly used as a metallization material, and copper metallization must be performed multiple times in the 400–450 °C range. In addition, water absorption is a major factor that degrades the electrical performance and reliability of low-Dk polymeric materials in electronic devices, i.e., it increases current leakage. Therefore, we aimed to develop a material with good dielectric properties, thermal stability, and low water absorption that can be utilized as insulating layers. In this study, we focused on the feasibility and expected effects of this molecular design of novel low-Dk FPI films containing an MCM-41-type amino-functionalized mesoporous silica (AMS) and its suitability as an insulating layer. Of course, dielectric properties can be supported by the behavior of charge carriers in the polymer analyzed through AC conductivity. Therefore, we plan to conduct future study on the molecular mechanisms that determine dielectric properties and kinetic properties by considering both dielectric properties through dielectric constant, dielectric loss and electrical properties through AC conductivity.

  1. corrections: line217 Figure 4 (not 6)

Author reply: We thank the reviewer for the valuable comment. Based on reviewer’s comment, we have modified the text as below.

Revised text [Manuscript, page 8]:

Figure 4(b)–(d) and Table S1 demonstrates the temperature dependence of viscoelastic behaviors of the FPI-g-AMS films.

  1. corrections also at lines 261-264 please clarify solubility S calculation.

Author reply: We thank the reviewer for the valuable comment. Below, we are providing detailed clarification for the solubility S calculation.

Revised text [Manuscript, page 8]:

where P° = Jd/p, P = Jsd/p = SD/p, J and Js denote the water vapor flux at time t at steady state, d is the sample thickness, p is the differential water vapor pressure, P is permeability, D is diffusivity, and S is solubility.

Reviewer 2 Report

Comments and Suggestions for Authors

Ref.comments to the paper titled as “ Frequency-Dependent Dielectric Permittivity and Water Permeability in Ordered Mesoporous Silica-Grafted Fluorinated  Polyimides” written by the authors: Jaemin Son, Hwon Park, Minju Kim, Jae Hui Park, Ki-Ho Nam, Jin-Seok Bae.

It is well known that the polyimide materials are very interesting and important as the effective matrix structure to create the organic and inorganic/organic systems. The authors of the current paper have developed the organic/inorganic hybrids composites by in-situ polymerization with the following investigation of their properties. The accent has been given on the study of the frequency-depended dielectric permittivity. From this point of view, the current paper is modern and actual.

For the first, this paper is good constructed and included the synthetic process, schematic models, instrumentation tools, good explanation of the properties of the novel composites.  Moreover 35 papers from literature database have been analyzed. Furthermore, the papers published last 5 years have been considered as well. Good!

Well. Materials and methods sections are prepared in details. Results and discussion part is interesting. I have seen the schematic illustration (Fig.1), the scheme of the synthesis process (Fig.2), SEM, FTIR, XRD experiments (Fig.3), TGA analysis (Fig.4), and frequency-dependent dielectric constant (Dk) and dissipation factor (Df) of the developped films at room temperature (Fig.5). Absorption and thermomechanical stability of the composites are shown as well.

Nice results! In this concern, please add or explain from these evidences the tendency to change the refractivity of the studied films. Is the refractive index can be changed dramatically for your hybrid films? It is important parameter, which is responsible for other materials features. It should correlate with the obtained results on dielectric permittivity.

Indeed, that is the recommendation for the authors. Basically, the researchers have own interesting approach and would not like to change anyone in the study.

Moreover, please extend the Conclusion section. Now it is so short and it is not included your basic results.

As for my general local opinion: The paper is interesting and prepared with good illustrations and discussion. It can be interesting in order to collect our knowledge in the materi9als science  and application. Moreover, it can be useful for the education process as well.

Thus, the paper can be published after minor corrections.

Author Response

Comments:

Ref.comments to the paper titled as “Frequency-Dependent Dielectric Permittivity and Water Permeability in Ordered Mesoporous Silica-Grafted Fluorinated Polyimides” written by the authors: Jaemin Son, Hwon Park, Minju Kim, Jae Hui Park, Ki-Ho Nam, Jin-Seok Bae.

It is well known that the polyimide materials are very interesting and important as the effective matrix structure to create the organic and inorganic/organic systems. The authors of the current paper have developed the organic/inorganic hybrids composites by in-situ polymerization with the following investigation of their properties. The accent has been given on the study of the frequency-depended dielectric permittivity. From this point of view, the current paper is modern and actual.

For the first, this paper is good constructed and included the synthetic process, schematic models, instrumentation tools, good explanation of the properties of the novel composites.  Moreover 35 papers from literature database have been analyzed. Furthermore, the papers published last 5 years have been considered as well. Good!

Well. Materials and methods sections are prepared in details. Results and discussion part is interesting. I have seen the schematic illustration (Fig.1), the scheme of the synthesis process (Fig.2), SEM, FTIR, XRD experiments (Fig.3), TGA analysis (Fig.4), and frequency-dependent dielectric constant (Dk) and dissipation factor (Df) of the developped films at room temperature (Fig.5). Absorption and thermomechanical stability of the composites are shown as well.

Author reply: We appreciate the reviewer’s valuable comment. Below, we are providing detailed clarification for the points raised by the reviewer. Here, we include our point-by-point responses to the specific comment below.

  1. Nice results! In this concern, please add or explain from these evidences the tendency to change the refractivity of the studied films. Is the refractive index can be changed dramatically for your hybrid films? It is important parameter, which is responsible for other materials features. It should correlate with the obtained results on dielectric permittivity.

Author reply: We thank the reviewer’s valuable comment. The refractive index is related to the permittivity according to Maxwell's equations as follows:

From this equation, it can be understood that the refractive index is proportional to the square root of the material's dielectric constant. Since the refractive index of silica is lower than that of polyimide, increasing the incorporation of silica will reduce the refractive index. Consequently, as the incorporation of silica increases, the refractive index of the hybrid film decreases, the dielectric constant can also be expected to decrease according to this relationship. In this study, it was observed that as the silica content increased, the dielectric constant decreased, which is anticipated to be a result of the decrease in the refractive index. In future study, we aim to elucidate the correlation between the dielectric properties and the polymer chemical structure by measuring the refractive index.

  1. Indeed, that is the recommendation for the authors. Basically, the researchers have own interesting approach and would not like to change anyone in the study.

Author reply: We thank the reviewer for the valuable comment. We confirmed that we will not change the researchers in this study.

  1. Moreover, please extend the Conclusion section. Now it is so short and it is not included your basic results.

Author reply: We thank the reviewer for the valuable commentary on how we could improve the manuscript. We have extended conclusions in the manuscript as follows.

Revised text [Manuscript, page 10]:

We successfully prepared novel low-Dk FPI films containing an MCM-41-type AMS, which exhibited an ultralow dielectric constant of ~2.69 and a very low dissipation factor of ~0.0026 at 1 MHz. The surface of the FPI-g-AMS films exhibited obvious hydrophobic characteristics that endow the films with outstanding water resistance, with FPI-g-AMS-3 showing the remarkably low water permeability of 1.10·10–8 mol s–1 m–1 atm–1. This could be attributed to the APTES moieties increasing the hydrophobicity of AMS surfaces, i.e., inhibiting the approaching of the water molecules and thus avoiding the hydrolysis of Si–O–Si bonds of the AMS pore walls. The increased tortuosity caused by the AMS particles also reduced the water permeability. In addition, the FPI-g-AMS films showed high thermooxidative/thermomechanical stability, including a high 5% weight loss temperature (>531 °C), char residue at 800 °C (>51%), and glass transition temperature (>300 °C). Therefore, the introduction of a hydrophobic cross-linked organic/inorganic hybrid network is beneficial to effectively reduce Dk, improve water resistance, and simultaneously maintain the overall physical properties of PIs. The proposed FPI-g-AMS films showed great potential in the applications of next-generation dielectric materials in microelectronic industry. 

  1. As for my general local opinion: The paper is interesting and prepared with good illustrations and discussion. It can be interesting in order to collect our knowledge in the materials science and application. Moreover, it can be useful for the education process as well. Thus, the paper can be published after minor corrections.

Author reply: We appreciate the reviewer’s valuable comment.

Reviewer 3 Report

Comments and Suggestions for Authors

In the present work, authors reported the synthesis of a series of novel low-Dk polyimide films containing an MCM-41-type amino-functionalized mesoporous silica via in-situ polymerization and subsequent thermal imidization, and then investigated their morphological, thermal properties, frequency-dependent dielectric behaviors, and water permeabilities. Results indicated that incorporating 6 wt.% AMS reduced the Dk at 1 MHz from 2.91 of the pristine fluorinated polyimide to 2.67 of the AMS-grafted FPI (FPI-g-AMS). Overall, this work was interesting and has certain reference function as an applied research. However, some issues should be addressed.

1, The authors should elaborate the general applicability of the current work, such as flexible printed circuit board, back plane?

2, Since authors have mentioned that polymers with low dielectric constant (Dk) are promising materials for high-speed communication networks, it is imperative to consider the frequency spectrum that is pertinent to such applications. For high-speed communication, the frequency range should extend beyond 0.45 GHz and potentially reach up to 50 GHz. It is curious that the investigation into the dielectric performance was conducted at a frequency of 1 MHz. The reason behind this choice of frequency is not immediately clear and requires clarification. Please give more details.

3, The chemical processing and characterization were well introduced. However, the successful preparation of PI was not well confirmed. It is suggested to carry out NMR analysis to confirm the successful preparation of PI and its relative composites. (optional)

4, Some key and important research results in PI field should be mentioned and cited so that we can provide a solid background and progress to the readers, such as Polymer Composites 2016, 37 (3), 907; Journal of Polymer Science 2022, 60(7), 1090.

5, The morphology images of composites were too small. Please replace them by another high-resolution ones. In addition, the particles of AMS were not visible from the images. Please explain this.

6, It was said that “However, the thermal degradation of the FPI-g-AMS films was accelerated above 600 °C, and char residue at 800 °C slightly decreased with the increasing AMS loading. It could be ascribed to the relatively higher thermal conductivity of the mesoporous silica compared to that of the FPI matrix.”. How the accelerated thermal degradation was relative to the higher thermal conductivity of the mesoporous silica? Please give more details? I believed the accelerated thermal degradation was ascribed to the relatively poor thermal stability of AMS as shown in Figure 1d.

7, For thin-film materials, particularly those utilized in high-speed communication networks, mechanical properties are as crucial as dielectric performance. Unfortunately, the current study lacks a comprehensive analysis of the polyimide thin-film’s mechanical properties, which is crucial for the material’s practical application.

8, It was said that “On the other hand, the increase in interfacial polarization due to the high concentration of AMS (>3 wt.%) suppresses the reduction effect of Dk in the FPI-g-AMS films. ”. The explanation should be considered again. Because the interfacial polarization was the main cause for the enhanced dielectric loss in such frequency range.

Author Response

Comments:

In the present work, authors reported the synthesis of a series of novel low-Dk polyimide films containing an MCM-41-type amino-functionalized mesoporous silica via in-situ polymerization and subsequent thermal imidization, and then investigated their morphological, thermal properties, frequency-dependent dielectric behaviors, and water permeabilities. Results indicated that incorporating 6 wt.% AMS reduced the Dk at 1 MHz from 2.91 of the pristine fluorinated polyimide to 2.67 of the AMS-grafted FPI (FPI-g-AMS). Overall, this work was interesting and has certain reference function as an applied research. However, some issues should be addressed.

Author reply: We appreciate the reviewer’s valuable comment. Below, we are providing detailed clarification for the points raised by the reviewer. Here, we include our point-by-point responses to the specific comments below.

  1. The authors should elaborate the general applicability of the current work, such as flexible printed circuit board, back plane?

Author reply: We thank the reviewer for the valuable commentary on how we could improve the manuscript. Based on reviewer’s comment, we have added the general applicability of our study as follows.

Revised text [Manuscript, page 1]:

Recently, the remarkable progress of electronic devices capable of high-speed data communication has further increased the demand for electronic substrates with low dielectric constant (Dk) and high electrical insulation properties [1–3]. Reducing the permittivity of materials with electrical insulation properties can improve signal transmission speed and efficiency, which is an effective way to promote the development of high-frequency and high-speed flexible circuit boards [4]. Aromatic polyimides (PIs) have been broadly used as insulating layers and electronic packages in the microelectronic industry and proposed as potential candidates for printed circuit boards, flexible display screen, and next-generation interlayer dielectric materials because of their excellent thermal stability, mechanical strength, dielectric properties, and chemical resistance [5,6].

Added references

[4] Liu, Y.; Zhao, X.Y.; Sun, Y.G.; Li, W.Z.; Zhang, X.S.; Luan, J. Synthesis and Applications of Low Dielectric Polyimide. Resour. Chem. Mater. 2023, 2, 49–62, doi:10.1016/j.recm.2022.08.001.

Revised text [Manuscript, page 10]:

Therefore, the introduction of a hydrophobic cross-linked organic/inorganic hybrid network is beneficial to effectively reduce Dk, improve water resistance, and simultaneously maintain the overall physical properties of PIs. The proposed FPI-g-AMS films showed great potential in the applications of next-generation dielectric materials in microelectronic industry. 

  1. Since authors have mentioned that polymers with low dielectric constant (Dk) are promising materials for high-speed communication networks, it is imperative to consider the frequency spectrum that is pertinent to such applications. For high-speed communication, the frequency range should extend beyond 0.45 GHz and potentially reach up to 50 GHz. It is curious that the investigation into the dielectric performance was conducted at a frequency of 1 MHz. The reason behind this choice of frequency is not immediately clear and requires clarification. Please give more details.

Author reply: We thank the reviewer’s valuable comment. In this study, FPI-g-AMS films were proposed to lower the dielectric constant. The well-defined mesoporous architecture in the FPI-g-AMS films considerably reduces the number of polarizing molecules per unit volume. In general, the dielectric constant of a material is dependent on frequency, and the dielectric constant tends to decrease as the frequency increases. However, the proposed FPI-g-AMS films may have interfacial polarization effects due to the disparity in dielectric response of FPI and AMS, which can also affect the dielectric properties. Interfacial polarization tends to happen slowly and becomes more prominent over extended time periods (low frequencies) and in the high frequency region, these interfacial and orientational polarizations are reduced. Thus, in this study, the dielectric properties were investigated up to 1 MHz frequency to consider not only the reduction in the number of polarized molecules per unit volume of the FPI-g-AMS films because of AMS addition, but also the interfacial polarization depending on AMS loading.

  1. The chemical processing and characterization were well introduced. However, the successful preparation of PI was not well confirmed. It is suggested to carry out NMR analysis to confirm the successful preparation of PI and its relative composites. (optional)

Author reply: We thank the reviewer’s valuable comment. The most common application of NMR spectroscopy is in analyzing substances in their liquid or solution state. And deuterium-substituted solvents such as CDCl3, DMSO-d6, etc are used as NMR solvent. However, the synthesized FPI-g-AMS films showed limited solubility in these solvents, so it is difficult to measure solution NMR. In addition, solid-state NMR spectroscopy is also used, but solid-state molecules have limited motion, so the peaks appear broad and unclear due to anisotropic interactions. The films in this study are difficult to be analyzed by such solid-state NMR due to the large number of monomers used and the complexity of their structure. In addition, successful preparation of FPI-g-AMS films was confirmed by FT-IR at lines 174–187. There are some reported papers confirming the structure of synthesized films without NMR analysis.

  1. Some key and important research results in PI field should be mentioned and cited so that we can provide a solid background and progress to the readers, such as Polymer Composites 2016, 37 (3), 907; Journal of Polymer Science 2022, 60(7), 1090.

Author reply: We thank the reviewer’s valuable comment. We have cited Polymer Composites 2016, 37 (3), 907 in the manuscript. Because the morphology behavior upon nanofiller addition observed by SEM in this paper is similar to the morphology behavior of the FPI-g-AMS films studied in our study.

Added references

[4] Liu, P.; Yao, Z.; Li, L.; Zhou, J. In Situ Synthesis and Mechanical, Thermal Properties of Polyimide Nanocomposite Film by Addition of Functionalized Graphene Oxide. Polym. Compos. 2016, 37, 907–914, doi:10.1002/pc.23249.

  1. The morphology images of composites were too small. Please replace them by another high-resolution ones. In addition, the particles of AMS were not visible from the images. Please explain this.

Author reply: We thank the reviewer’s valuable comment. Based on reviewer’s comment, we are added high-resolution SEM images as follows.

Revised Figure caption [Manuscript, page (6)]:

Figure 3. (a) Fourier-transform infrared spectra and (b) XRD patterns of the FPI-g-AMS films. (c) low magnification and (d) high magnification SEM fracture surface morphologies of the FPI-g-AMS films. (e) Elemental mapping images of the FPI-g-AMS-3 film.

  1. It was said that “However, the thermal degradation of the FPI-g-AMS films was accelerated above 600 °C, and char residue at 800 °C slightly decreased with the increasing AMS loading. It could be ascribed to the relatively higher thermal conductivity of the mesoporous silica compared to that of the FPI matrix.”. How the accelerated thermal degradation was relative to the higher thermal conductivity of the mesoporous silica? Please give more details? I believed the accelerated thermal degradation was ascribed to the relatively poor thermal stability of AMS as shown in Figure 1d.

Author reply: We thank the reviewer’s valuable comment. The thermal conductivity of mesoporous silica is significantly higher than that of the FPI matrix. With increasing loading of AMS, which possesses superior thermal conductivity, a continuous thermally conductive network is established within the polymer matrix. This enhanced heat transfer by AMS facilitates more uniform heating throughout the matrix, resulting in more homogeneous thermal decomposition of the material. Consequently, compared to lower AMS loadings, the char content may be reduced.

  1. For thin-film materials, particularly those utilized in high-speed communication networks, mechanical properties are as crucial as dielectric performance. Unfortunately, the current study lacks a comprehensive analysis of the polyimide thin-film’s mechanical properties, which is crucial for the material’s practical application.

Author reply: We thank the reviewer’s valuable comment. Dynamic mechanical analysis (DMA) was used to determine dynamic mechanical properties of the polyimide films. DMA analyzes viscoelastic behavior by applying oscillating strain to a sample and measuring the resulting stress, or by applying periodic stress and measuring the resulting strain. The storage modulus and loss modulus obtained by DMA are related to the stresses in the molecular chains. The storage modulus indicates elastic solid behaviors, and a higher storage modulus indicates a stronger resistance to external deformation. In addition, a material for an intermetal dielectric requires a storage modulus of over 1 GPa. And Kapton®, a commercial polyimide film used as an insulating layer in the microelectronic industry, which shows a storage modulus of 2 GPa. In comparison to Kapton®, all prepared films also showed a storage modulus of over 2 GPa up to 290 ℃, meeting the requirements of the insulating layer. Thus, it can be considered that the films have sufficiently high mechanical properties. It is expected that DMA can be used to characterize the mechanical properties of the films.

  1. It was said that “On the other hand, the increase in interfacial polarization due to the high concentration of AMS (>3 wt.%) suppresses the reduction effect of Dk in the FPI-g-AMS films. ”. The explanation should be considered again. Because the interfacial polarization was the main cause for the enhanced dielectric loss in such frequency range.

Author reply: We thank the reviewer’s valuable comment. The difference in conductivity and dielectric constant between the inorganic AMS and organic FPI matrix induces interfacial polarization effect. This results in a strong local electric field at the interface, capturing free electrons generated under applied electric fields. The high electric field strength causes charge accumulation at the interface, leading to an increased dielectric constant in the FPI-g-AMS films due to enhanced spatial charge. Increasing the amount of AMS increases interface presence in the hybrid film, bringing AMS particles closer together and reinforcing the local electric field, thereby increasing interfacial polarization. As described at line 240, the well-defined mesoporous architecture in the FPI-g-AMS films significantly reduces the number of polarizing molecules per unit volume. However, the addition of more than 3 wt% AMS reduces the magnitude of the dielectric constant decrease due to the increase in interfacial polarization. In other words, we found that the dielectric constant does not continue to decrease with increasing AMS content. This is also true for dielectric loss, where the dielectric loss decreased with increasing AMS content when AMS was added up to 3 wt%. However, when 6 wt% AMS was added, the dielectric loss increased in such frequency range due to the increase in interfacial polarization.
